# Limitations of refinement methods for weak to strong generalization

**Seamus Somerstep**[*][♠] **Ya'acov Ritov**[♠] **Mikhail Yurochkin**[★]
**Subha Maity**[▲] **Yuekai Sun**[♠]
♠ University of Michigan  ★ IFM MBZUAI  ▲ University of Waterloo

## Abstract

Standard techniques for aligning large language models (LLMs) utilize human-produced data, which could limit the capability of any aligned LLM to human level. Label refinement and weak training have emerged as promising strategies to address this *superalignment* problem. In this work, we adopt probabilistic assumptions commonly used to study label refinement and analyze whether refinement can be outperformed by alternative approaches, including computationally intractable oracle methods. We show that both weak training and label refinement suffer from irreducible error, leaving a performance gap between label refinement and the oracle. These results motivate future research into developing alternative methods for weak to strong generalization that synthesize the practicality of label refinement or weak training and the optimality of the oracle procedure.

## 1 Introduction

Given the rapid advancement of LLMs, it is important to ensure that they align themselves with human values. Standard techniques for doing so, such as supervised fine-tuning and reinforcement learning from human feedback (Ouyang et al., 2022) utilize human-produced data, which could limit the capability of any aligned LLM to human level. This has led to a continued interest in developing techniques for aligning a super-human model; such a goal is referred to as the *superalignment of large language models*, a term introduced by OpenAI (Leike and Sutskever, 2023).

The same team introduced *weak to strong generalization* as an analogy for superalignment (Burns et al., 2023). Weak to strong generalization is meant to act as an empirical verification framework for superalignment techniques: a small LLM acts as a human producing data while a large LLM acts as the superhuman model. Two promising classes of techniques for superalignment/weak to strong generalization have emerged. The first directly utilizes the weak data in an alignment/training process; this includes direct weak training and bootstrapping chains of models with weak training (Burns et al., 2023). Understanding the success of this class of techniques is a work in progress; the authors of (Burns et al., 2023) find empirical success for them, and in some theoretical frameworks, these techniques are provably successful (Lang et al., 2024; Charikar et al., 2024; Wu and Sahai, 2024; Xue et al., 2025). On the other hand, weak training can lead to deception of the strong model and reduce its capabilities in tasks such as reasoning (Yang et al., 2024b;a) (*cf.* Figure 1 for a replication). Additionally, in the theoretical

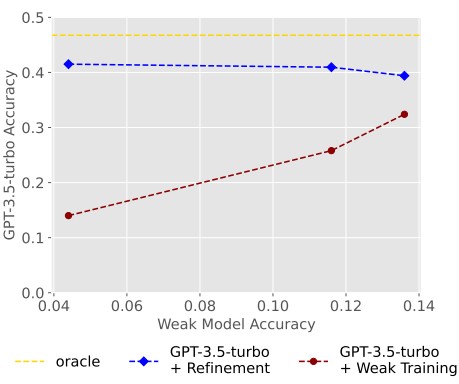

Figure 1: Performance of weak-to-strong methods on GSM8K. Details are provided in Appendix B.

---

*smrstep@umich.edu

frameworks proposed in (Somerstep et al., 2024; Yao et al., 2025) training on the weak model outputs can actually degrade strong model capabilities. In response to this uncertainty, a second class of weak to strong generalization techniques that utilize latent capabilities of the large pre-trained LLM to refine the weak labels (Somerstep et al., 2024; Yang et al., 2024b) has emerged. While the empirical results of these techniques are promising, and some theory underpins their success, it is still unknown if refinement (or weak training) is the best that we can do in weak to strong generalization settings. The authors of Yang et al. (2024b) observe that, empirically, refinement underperforms training on gold standard target data (in this context data produced by a capable LLM), a phenomenon that we also observe in Figure 1. Inspired by this, we seek to study if refinement (or weak training) is optimal in certain settings.

To get a partial answer, we adopt a probabilistic set-up (a generalization of Somerstep et al. (2024)) used to justify refinement, and study if it (and direct weak training) is optimal for weak to strong generalization. Our framework is a *transfer learning* framework, the source distribution $(X, Y) \sim P$ represents data from the unaligned model while $(X, Y) \sim Q$ represents data from an aligned version of the strong model. The key challenge in weak to strong generalization is that we only observe weak labels $(Y')$ over $Q$. From the source and weak data, the goal is to obtain an estimate of the distribution $Q_{Y|X}$.

| superalignment | weakly-supervised TL |
|---|---|
| pretrained LLM | $Y_P \mid X_P$ |
| (super)alignment task | $Y_Q \mid X_Q$ |
| weak teacher | $Y'_Q \mid X_Q$ |

Table 1: Transfer Learning $\longleftrightarrow$ Superalignment

The difference between the base and hypothetical fine-tuned version of an LLM is complex; we assume that this differnce may be attributed to a *latent concept shift*. Latent concept shift occurs when both the source and target distributions are mixtures of distributions indexed by an unobserved latent concept variable $K$, with only the proportions of $K$ changing between $P$ and $Q$.

Following Wu and Sahai (2024), we consider two desiderata that ensure a (non-trivial) weak to strong generalization has occurred:

1. The weak to strong estimator should asymptotically converge to the target (as the number of weak target and source samples grows).

2. Estimators trained with either only the weak target data or the source data should not asymptotically converge to the target task.

The second requirement lays a framework where a weak to strong generalization task is not trivial. Our primary contribution is summarized in the following (informal) theorem.

**Theorem 1.1.** *Under the latent concept shift framework, both source and (weak) target distributions are required to produce a consistent estimator. Additionally, the following holds:*

*1. Both label refinement and weak training produce inconsistent estimators of the target function.*

*2. There exists a deconvolution-based procedure meets the consistency criteria.*

The remaining discussion sets up the latent concept shift framework and provides results demonstrating when 1.1 holds. As further details about the deconvolution-based procedure are deferred to Section 4, for now, we'd like to note that it's an idealized procedure and may not be practical for actual weak to strong generalization tasks (e.g. mathematical reasoning); rather, we develop it to show that there is a non-trivial gap in current techniques and demonstrate possible room for improvement in weak to strong generalization tasks.

## 1.1 Related work

**Transfer Learning:** In transfer learning, the goal is to leverage a pre-trained model or an abundant source of data to improve model performance on a target task (as compared to only training on any target data). Often, the form of knowledge transfer that must occur is categorized by the difference in the source and target distributions. Generally, the three categories of transfer are covariate shift ($P(X) \neq Q(X)$) (Kpotufe and Martinet, 2018; Huang et al., 2006; Dai et al., 2007), label shift $P(Y) \neq Q(Y)$ (Maity et al., 2020; Lipton et al., 2018; Zhang et al., 2015), and posterior drift ($P(Y|X) \neq Q(Y|X)$)(Maity et al., 2021; Cai and Wei, 2019; Liu et al., 2020). Our work falls in the category of posterior drift; as in prior works, we must make assumptions on how the source function $\mathbb{E}_P(Y|X)$ and $\mathbb{E}_Q(Y|X)$ relate. In Cai and Wei (2019); Cai and Pu (2024); Maity et al. (2021) the difference must lie in a simple function class (*e.g.*, a linear function of a sufficient statistic or a polynomial class). In our work, as in Alabdulmohsin et al. (2023), the posterior drift is due to distribution shift in an unobserved random variable.

**Weak to Strong Generalization/Superalignment:** Several prior works have conducted theoretical studies of weak to strong generalization (Lang et al., 2024; Charikar et al., 2024; Wu and Sahai, 2024; Somerstep et al., 2024). The closest related work is Somerstep et al. (2024) which considers a similar causal graph but restricts the relationship between $X, Y$ and $Y'$ to a mixture of linear regressions. In contrast with this work and the work in Somerstep et al. (2024), others have considered frameworks where weak training is sufficient for weak to strong generalization (Lang et al., 2024; Charikar et al., 2024; Wu and Sahai, 2024). Wu and Sahai (2024) show that weak training allows for weak to strong generalization in spiked covariance models through a benign overfitting process. Charikar et al. (2024), show that weak training may lead to good generalization properties if the base and target models are close with respect to the representation layer. Lang et al. (2024); Shin et al. (2024) studies properties of the weak data, and establish desirable conditions for weak training.

Interest in the weak to strong generalization problem has grown on the empirical side. One line of work transfers alignment behavior induced by RLHF from a weak model to a strong model by learning the distributional differences in the weak model before and after alignment (Zhu et al., 2024; Zhou et al., 2024). Other works look to expand generalization from weak training to other model properties beyond accuracy. For example, the authors of Yang et al. (2024b) improve large models' mathematical reasoning with responses from smaller models while in Pawelczyk et al. (2024) trustworthiness properties such as fairness, privacy, and adversarial robustness are transferred.

Other lines of work include leveraging the temperature parameter for weak training (Zhang et al., 2024), studying deception from weak training (Yang et al., 2024a), and using a dynamic logit fusion approach for weak training (Fan et al., 2024).

## 2 Set Up: Latent concept transfer

Let $P$ and $Q$ be the data distributions in source and target domains with corresponding probability distribution function (or probability mass function) $p$ and $q$. Over the source and target distributions, there are three *observable* random variables of interest denoted $X$, $Y, Y'$. The variable $X$ generally denotes a model input, $Y$ is a gold-standard/uncorrupted model output, and $Y'$ are proxy labels produced from a smaller model.

**Goal:** We wish to learn the target predictive distribution $q(y|x)$ from knowledge of $p(x, y, y')$ and $q(x, y')$.

To make progress, we begin with specifying the conditional dependencies of the data-generating process.

**Assumption 2.1.** *All data drawn from P or Q is generated by the process given in Figure 2 and this process is faithful and Markov. Additionally, K is never observed in either P or Q and Y is never observed in Q.*

Concretely, the data generating process in the source and target is specified as a probabilistic directed acyclic graph. The Markovian and faithfulness assumptions simply imply that conditional independences exist in the data generating process iff they exist in the graph. This factorizes the source and target distributions as

$$p(x, k, y, y') = p(x)p(k|x)p(y \mid x, k)p(y' \mid x, k),$$
$$q(x, k, y, y') = q(x)q(k|x)q(y \mid x, k)q(y' \mid x, k).$$

Table 1 summarizes the analogy between transfer learning and weak-to-strong generalization. The variable $X$ plays the role of a natural language query to a small or large LLM, $K$ is a latent mechanism through which models generate an output, $Y$ are outputs from a strong model, and $Y'$ are outputs from a weaker model. As in other transfer learning problems, the learner has access to source and target samples, though it is key to note that the true labels $Y$ *are only observed over P*.

**Assumption 2.2.** *We assume that the learner has sampling access to the following distributions* $(X, Y, Y') \sim P$ *and samples* $(X, Y') \sim Q$*. The latent variable K is never observed.*

In most weak to strong generalization applications, the learner always has access to queries from some dataset of interest, and a base/unaligned large LLM (*e.g.* a base version of Llama2-70b). The sampling access to $(X, Y)$ from $P$ represents this. Likewise, access to a weak/small LLM with some expertise on the target task (*e.g.* a base version of Llama2-7b fine-tuned on good data) is always given; this is represented by the sampling access to $(X, Y')$ over $Q$.

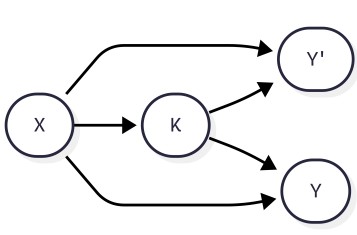

Figure 2: Latent concept transfer

Access to $Y'$ over $P$ depends on the specific weak to strong generalization setting. In some cases, direct access to the base/unaligned version of the small LLM (e.g. an unaltered version of Llama2-7b) and observing the difference between the unaligned and aligned small LLM can be used to "reconstruct" the aligned large LLM (Zhu et al., 2024; Zhou et al., 2024). In other works, namely Somerstep et al. (2024); Yang et al. (2024b), base small LLM observations are not available (i.e. $(X, Y')_P$ is not physically observable data); rather, the conditional distribution $P(Y|X, Y')$ represents the outcome distribution of the base model fed with an $X$ and weak label $Y'$. We primarily work in this setting.

In latent concept shift, the key assumption is that the shift between source and target occurs in the frequency of $K$. We also allow for a shift in $Y'|X$, our reasoning is explained below.

**Assumption 2.3.** *We assume that* $p(x) = q(x)$*, and* $p(y|x, k) = q(y|x, k)$*, but* $p(k|x) \neq q(k|x)$ *and* $p(y'|x, k) \neq q(y'|x, k)$*.*

The shift from $p(k|x)$ to $q(k|x)$ occurs due to the shift in prior over $K$ from $P$ to $Q$. The shift between $p(y'|x, k)$ and $q(y'|x, k)$ represents some imperfection in the aligning process of the weak model. In our setup, recall that $p(y|x, y')$ represents the output distribution of an LLM fed with an input $x$ and a weak label $y'$; it is not reasonable to expect that the base LLM is aware of the exact bias present in $y'$. This is encoded by appropriately restricting the shift between $p(y'|x, k)$ and $q(y'|x, k)$. Note that we assume the target version of the strong model has (conditional on $k$) density $p(y|x, k)$; or that this imperfection is absent in the hypothetical gold standard model. The final assumption we make parameterizes each of the distributions of interest.

**Assumption 2.4.** *Let* $\pi$ *represent the marginal distribution of the latent concept. Then for known density functions* $g(\cdot)$ *and* $\varphi(\cdot)$ *parameterized by* $\eta$ *and* $\theta$ *respectively, the source and target distributions satisfy*

$$p(y|x) = \sum_k \frac{\pi_k^p g(x|\eta_k)}{\sum_{k'} \pi_{k'}^p g(x|\eta_{k'})} \varphi(y; x, \theta_k); p(y'|x) = \sum_k \frac{\pi_k^p g(x|\eta_k)}{\sum_{k'} \pi_{k'}^p g(x|\eta_{k'})} \varphi(y'; x, \theta_k^{w_p}) \quad (2.1)$$

$$q(y|x) = \sum_k \frac{\pi_k^q g(x|\eta_k)}{\sum_{k'} \pi_{k'}^q g(x|\eta_{k'})} \varphi(y; x, \theta_k); q(y'|x) = \sum_k \frac{\pi_k^q g(x|\eta_k)}{\sum_{k'} \pi_{k'}^q g(x|\eta_{k'})} \varphi(y'; x, \theta_k^{w_q}). \quad (2.2)$$

Assumption 2.4 parametrizes the relationship between $X$, $Y$, $Y'$, and $K$. Our parametric assumption 2.4 leaves questions such as identifiability novel, and is general enough to cover prevalent examples in the literature.

**Example 2.5.** *Wang et al. (2024) assumes that there is a set of language modeling tasks $\mathcal{T}$ indexed by $d$, and that for the d-th task it holds that $y = f(x, \theta^d, \epsilon)$, $x \sim P_X$, $\epsilon \sim P_\epsilon$. By equating the d-th task with the k-th value of the latent concept in our setup, we see that the framework of Wang et al. (2024) is covered under Assumption 2.4.*

**Example 2.6.** *Somerstep et al. (2024) assume that $y = \sum_k \pi_k \beta_k^\top x + \epsilon$, with $\epsilon \sim \mathcal{N}(0, \sigma^2)$. Pathak et al. (2024) show that there exists a transformer architecture that matches this mixture of regression distributions.*

**Example 2.7.** *Xie et al. (2021) posit that an LLM can be represented by a hidden Markov model (HMM) mixture. In particular, there is a set of transition matrices $\Theta = \{\theta_1, \ldots, \theta_K\}$, priors $\pi^p$, $\pi^q$, emission parameters $\tau$, $\tau^w$, and a state space $\mathcal{H}$ such that the sequence of tokens $(x, y) = (x^1, x^2, \ldots, x^l, y)$ satisfies*

$$p(y|x) = \sum_k \left[ \sum_{h \in \mathcal{H}} p(y|h, \tau) p(h|x, \theta_k) \prod_{j=1}^{l} \sum_{h \in \mathcal{H}} p(x^j|h, \tau) p(h|x_{<j}, \theta_k) \right] \pi_k^p$$

$$q(y'|x) = \sum_k \left[ \sum_{h \in \mathcal{H}} p(y'|h, \tau^w) p(h|x, \theta_k) \prod_{j=1}^{l} \sum_{h \in \mathcal{H}} p(x^j|h, \tau) p(h|x_{<j}, \theta_k) \right] \pi_k^q$$

We return to the motivation of problem set up: recall $X$ are meant to represent queries to an LLM, while $Y/Y'$ represent outputs of large and small language models. The latent variable represents some internal mechanism by which language models produce their responses. In each of these examples it assumed that *language models are essentially emulating the distributions they are trained on.* In other words, we are assuming that the base (strong model) is trained on a distribution specified by equation (2.1) while the target model is an LLM (hypothetically) trained on a distribution specified by equation (2.2). When an LLM receives a query, it internally attempts to infer the concept most related, then samples a response from the corresponding expert.

## 3 Weak to strong generalization strategies

In this section we discuss three possible approaches to the weak to strong generalization problem and their performance within our framework. The first two approaches, weak training and label refinement, have been previously proposed and studied (Burns et al., 2023; Somerstep et al., 2024; Yang et al., 2024b). Within our framework, we will see that each of these fails to produce estimators that satisfy the weak to strong generalization desiderata. Inspired by this, we propose a new weak to strong generalization procedure, which is impractical to implement but does enjoy superior theoretical properties.

### 3.1 Weak Training

Burns et al. (2023) advocate for training a base model, then training the base model for a small number of epochs on the weak data.. Consider the regression settings of Examples 2.5 and 2.6. In our framework, this essentially corresponds to fitting an estimator of the form

$$\hat{\pi}_{\text{wt}}, \hat{\theta}_{\text{wt}} = \arg \min_{\theta \in \Theta, \pi \in \Delta(K)} \hat{\mathcal{L}}_{wt}(\theta),$$

$$\hat{\mathcal{L}}_{wt}(\theta, \pi) \triangleq \sum_{i=1}^{n_{Q'}} \left\{ \sum_k \pi_k f(x_i, \theta_k) - y_i' \right\}^2 + \lambda \sum_{i=1}^{n_p} \left\{ \sum_k \pi_k f(x_i, \theta_k) - y_i \right\}^2.$$

Here $\lambda > 0$ controls how much attention the estimator should pay to the source versus the weak target data. Fitting an estimator to a loss combining both sources of data is a core component of many transfer learning techniques Maity et al. (2020); Cai and Wei (2019); unfortunately, the presence of weak supervision in the target domain prevents this from being particularly useful in our setting.

**Proposition 3.1.** *Consider the regression setting of examples 2.5 and 2.6 with $n_P = n'_Q \triangleq n$. Let $\epsilon_P, \epsilon_{Q'}$ be the $K \times 1$ vectors of the bias in the source and weak models conditioned on $k$. Suppose that $\mu(x; \theta)$ is Lipschitz in $\theta$. Then it holds that $\hat{\theta}_{wt} \xrightarrow{p} \theta^*_{wt}, \hat{\pi}_{wt} \xrightarrow{p} \pi^*_{wt}$ where $\theta^*_{wt}, \pi^*_{wt}$ satisfies*

$$\mathbb{E}_x\left[\left\{\mathbb{E}_Q[y|x] - \sum_k \pi^*_{wtk} f(x; \theta^*_{wtk})\right\}^2\right] \geq \eta ||\epsilon_P||^2 + (1-\eta)^2 ||\epsilon_{Q'}||^2 + \eta(1-\eta)\epsilon_P^\top \epsilon_{Q'}.$$

Proposition 3.1 shows that within our framework, weak training does not satisfy the first criteria of weak to strong generalization (it does not produce a consistent estimator of the target function). This inconsistency within our framework is also prevalent empirically; for example, in Figure 1 GPT-3.5-Turbo does not successfully learn any mathematical reasoning ability from the weak models. While Proposition 3.1 covers the regression case, the intuition carries over to the HMM case. Both the source and weak target data are biased, and thus constructing a consistent estimator by mixing the two is generally not possible.

## 3.2 Label Refinement

Simultaneously, Somerstep et al. (2024) and Yang et al. (2024b) introduced a set of methods which feed the weakly labelled data as auxiliary information to the source model and then *drawing a new label for each query in the target data set*. For example, in both Somerstep et al. (2024) and Yang et al. (2024b) the first step of the refinement procedure is to draw new labels $\hat{Y}$ that satisfy

$$\mathbb{P}(\hat{Y} = y|X = x) \stackrel{d}{=} \mathbb{E}_V \mathbb{P}_P(Y = y|X = x, V); \quad V \sim \prod_{j=1}^M Q_{Y'|X_j}. \tag{3.1}$$

In other words, for each query $x$, they collect $m$ weakly labelled pairs $(x_j, y'_j)_{j=1}^m$ drawn from $Q_{X,Y'}$ (*i.e.* produced by the weak model), form "auxiliary information" $v = [x_1, y'_1, \ldots, x_m, y'_m]$ and then draw a new label from the source model that is fed with both $v$ and $x$. The notation $\mathbb{P}_P(\cdot)$ emphasizes that the refined labels are being drawn from the source distribution.

Following Somerstep et al. (2024); Yang et al. (2024b) we utilize our framework to study refined data $(X, \hat{Y})$ with the following distribution:

$$X \sim Q_X; \quad p(\hat{y}|x) \triangleq q_{re}(y \mid x) = \int_V p(y \mid x, v) dQ(v \mid x).$$

It is also possible to perform label refinement by correcting one weak label at a time (Somerstep et al., 2024), in this case we consider a refined distribution of the form $q_{re}(y \mid x) = \int p(y \mid x, y') dQ(y' \mid x)$

### 3.2.1 Irreducible error of label refinement

In this subsection, we ask whether for a given query $x$, the refined label is a good stand-in for a hypothetical gold-standard label drawn from the target distribution. To begin, we work $q_{re}(y|x)$ into a more interpretable form.

**Proposition 3.2.** *For $\mathbf{P}\{k \mid X, k'\} = \int_V p(k \mid x, v) q(v|x, k') dv$, which is an entry of the (conditional-on-X) confusion matrix of the prediction $k$ (of true class $k'$ from $V$), one can write*

$$q_{re}(y|x) = \sum_k p(y|x, k)\hat{q}(k|x) \quad and \quad \hat{q}(k|x) = \sum_{k'} q(k'|x)\mathbf{P}\{k \mid X, k'\}.$$

For the case of correcting one weak label at a time, one can simply replace $v$ with $y'$.

Proposition 3.2 reveals the statistical intuition for refinement methods in weak to strong generalization: The model internally updates its prior on the latent variable when fed the weak data as auxiliary information. This essentially transfers the problem of estimating $q(x)$ from weak data to inferring the marginal of the latent variable $K$ from pairs $(X, V)/(X, Y')$. Unfortunately, Proposition 3.2 also reveals that even refinement does not produce labels that match hypothetical gold standard labels, as if $\hat{q}(k|x) \neq q(k|x)$ then $q_{re}(y|x) \neq q(y|x)$.

**Example 3.3** (ICL Refinement). *Consider the HMM set-up of Example 2.7, with $V \sim \prod_{j=1}^{M} Q_{Y'|X_j}$ and taking $q(k) = \mathbf{1}\{k = k^*\}$ for simplicity. We may then write*

$$\hat{q}(k|x) \propto \mathbb{E}_V p(k, x, v) \approx \pi_k^p \prod_{j=1}^{M} \mathbb{E}_{Q_{y'|x_j}} \Big[ \sum_h p(y'|\tau, h) p(h|x, \theta_k) p(x|\theta_k)) \Big] \qquad (3.2)$$

*The approximate treatment of the ICL pairs as* iid *is justified by the use of additional delimiter tokens in $V$. For simplicity, we take this as a given and refer to Xie et al. (2021) for a detailed discussion.*

*We also see that for a given $k \neq k^*$, $\hat{q}(k|x) > 0$ unless the expected likelihood of the sequence of tokens $x_1, y_1', \ldots, x_m, y_m'$ is zero under the transition matrix $\theta_k$. Thus, under general conditions, the labels $\hat{y}$ will still incur some bias from weighting towards the incorrect concepts, leading to irreducible error.*

**Example 3.4** (Weak label improvement). *The authors of Somerstep et al. (2024) also propose a refinement procedure in which the source model "corrects" the weak label for each data pair $(x, y')$. We return to the regression setting (Example 2.5) and again assume that $q(k) = \mathbf{1}\{k = k^*\}$. We can model this refinement step as*

$$\hat{q}(k|x) = \int_{y'} p(k \mid x, y') q(y'|x, k^*) dy'$$

*In Appendix A we show that if we let $\Delta_k^2(x) = (\mu(x; \theta_k^{w_p}) - \mu(x; \theta_{k^*}^w))^2$, then it holds that*

$$\hat{q}(k|x) \lesssim p(k|x) e^{-c\Delta_k^2(x)}.$$

We see that for $x$ in which $\Delta_k^2(x)$ is large for all $k \neq k^*$, the label bias can be drastically reduced. If, on the other hand, it turns out that $\Delta_{k^*}^2(x)$ is large due to the distribution shift in $y'|x$, then label refinement will suffer from a large bias and still have undesirable irreducible error. These examples are evident in Figure 2, where we see that label refinement improves on weak training, but does not compare to an oracle proxy.

### 3.3 Weak to strong generalization through latent concept identification

We have seen that within the latent concept transfer framework, both weak training and label refinement produce biased estimates of the target function. The question remains if there exists a procedure that produces a consistent estimate of the target function. To answer this in the affirmative, we propose a two-step procedure: first identify the latent mixture components and mixture proportions in $P$ and $Q$, then construct the target function by borrowing mixture proportions from $Q$ and the mixture components from $P$. We lay this out in Algorithm 1.

---

**Algorithm 1** Latent Concept Identification

---

**Require:** Source data $\mathcal{D}_P : \{x_i, y_i, y_i'\}$ target data $\mathcal{D}_{Q'} : \{x_i, y_i'\}$, maximum likelihood estimation procedure MLE
  1: Compute $\{\hat{\theta}_k, \hat{\theta}_k^{w_p}, \hat{\eta}_k, \hat{\pi}_k^p\}_{k=1}^K \leftarrow \text{MLE}(\mathcal{D}_P)$
  2: Compute $\{\hat{\theta}_k^{w_q}, \hat{\pi}_k^q\}_{k=1}^K \leftarrow \text{MLE}(\mathcal{D}_{Q'})$
  3: Compute assignment $a(k) = \arg\min_{k'} d^2(\hat{\theta}_k^{w_p}, \hat{\theta}_k^{w_q})$
  4: Compute final estimate $\hat{q}(x) \leftarrow \sum_k \hat{\pi}_{a(k)}^q g(x|\hat{\eta}_{a(k)}) / [\sum_{k'} \hat{\pi}_{a(k')}^q g(x|\hat{\eta}_{a(k')})] \varphi(y; x, \hat{\theta}_k)$

---

The success of this strategy hinges on the MLE/identification step, we discuss this further in Section 4. We briefly comment on possible relaxation on observing $Y'$ over $P$ for the success of Algorithm 1. In particular, note that $\hat{\theta}_k^{w_p}$ does not play a role in the final estimate; rather it is only used to compute the assignments. With no weak source samples available, one could instead opt to utilize $\theta_k$ in the assignment step.

## 4 Identification and estimation under latent concept shift and weak supervision

In this section, we study the question: *When are the parameters of $Q_{Y|X}$ identifiable?* To formalize identifiability, consider the space of measures $\Lambda(\mathcal{X} \times \mathcal{Y})$, and the space of measures on $\Lambda(\mathcal{X} \times \mathcal{Y})$, which we denote as $\Lambda^2(\mathcal{X} \times \mathcal{Y})$. For any mixture density specified by $f(y, x; \pi, \eta, \theta, K) \equiv \sum_{k=1}^{K} \frac{\pi_k g(x|\eta_k)}{\sum_{k'=1}^{K} \pi_{k'} g(x|\eta_{k'})} \varphi(y; x, \theta_k)$, we can define the *mixing measure* $F_*(\eta, \theta, K) \in \Lambda^2(\mathcal{X} \times \mathcal{Y})$ as the weighted sum of Dirac measures $F_*(\eta, \theta) = \sum_{k=1}^{K} \pi_k \delta_{\eta_k, \theta_k}$. As in prior work (Aragam et al., 2018), we say that a mixture family $f(x; \vec{\eta}, \vec{\theta})$ is identifiable if the map

$$\Pi : \Lambda(\mathcal{X} \times \mathbf{R}) \to \Lambda^2(\mathcal{X} \times \mathbf{R}) : \quad \Pi(f(x; \pi, \eta, \theta)) \to F_*(\eta, \theta)$$

is injective. Recall our goal is learning $q(y|x) = \sum_k \frac{\pi_k^q g(x|\eta_k)}{\sum_{k'} \pi_{k'}^q g(x|\eta_{k'})} \varphi(y; x, \theta_k)$. The necessary ingredients to identifying $q(y|x)$ are:

1. Identifiability of the mixture system. Note this property is dependent on the functions $\varphi(\cdot, \cdot, \cdot)$ and $g(\cdot, \cdot)$.

2. A matching identifiability condition between $\{\theta_k^{w_p}\}$ and $\{\theta_k^{w_q}\}$.

Our first contribution is to characterize the identifiability of our general mixture model; this result generalizes those in Nguyen et al. (2024b).

**Definition 4.1.** *We say that the functions $\varphi(\cdot, \cdot, \cdot)$ $g(\cdot, \cdot)$ are zero'th order algebraically independent if for any collection of distinct parameters $\{\theta_k\}$, $\{\eta_k\}$, the collection of functions in $x$ $\{g(x; \eta_k)\}$ is linearly independent and the collection of functions in $\{\varphi(y; x, \theta_k)\}$ in $x, y$ is linearly independent.*

**Proposition 4.2.** *Suppose that $\varphi(\cdot, \cdot, \cdot)$, and $g(\cdot, \cdot)$ are (zero'th order) algebraically independent. Then the mixture family $f(x; \vec{\eta}, \vec{\theta})$ is identifiable.*

Proposition 4.2 establishes a sufficient condition for the parameters $\vec{\eta}, \vec{\theta}, \vec{\pi}$ being identifiable from observing data $(X, Y/Y')$ from $P$ or $Q$. The algebraic independence condition ensures that distinct sets of the parameters cannot lead to the same distribution over $P$ and $Q$.

**Example 4.3.** *In Example 2.5, the density of $f(x, \theta, \epsilon)$ needs to satisfy algebraic independence; in Example 2.6 it is sufficient that the $\{\beta_k\}$ be linearly independent.*

In the case of Example 2.7, a discussion on when the necessary algebraic independence condition is met is more complex.

**Proposition 4.4.** *Consider the HMM model setup of Example 2.7 and in Xie et al. (2021). Following their notation, suppose some additional assumptions hold:*

1. *The state space $\mathcal{H}$ is finite, the tokens $x^j$ and $y$ also lie in a finite token space $\mathcal{O}$. Emission sequences $x^1, x^2, \ldots$ from the mixture of hidden markov models may be arbitrarily long.*

2. *For value $\tau$, there exists set of tokens $(o_1^*, \ldots, o_H^*)$, $o_h^* \in \mathcal{O}$ such that $p(o_h^*|h, \tau) > 0$ and $p(o_{h'}^*|h, \tau) = 0$.*

3. *$p(y; h, \tau)$ satisfies the algebraic independence condition in $\tau$.*

*Then $q(x, y')$ and $p(x, y)$ satisfy the algebraic independence conditions in $\theta$, $\pi$, and $\tau$.*

Assumption 2 of Proposition 4.4 is fairly strong; in the topic model literature, the special tokens $o_h^*$ are referred to as *anchor words* (Arora et al., 2012) and in hyperspectral mixing they are referred to as *pure pixels* (Jasmine and Pattabiraman, 2018). Anchor words play an important role in the identification of the parameters of topic models.

Unfortunately, because we do not observe $X, Y \sim Q$, to identify $q(y|x)$ we require more than just identifiability of the individual components. At its heart, we need to match the

target prior $\pi^q$ observed from $Q'$ to the source components $\eta, \theta$ observed from $P$, keeping in mind that each of these is identifiable only up to permutation. To assure this is possible, we impose a statistical distinguishability condition on $p(y'|x,k)$ and $q(y'|x,k)$.

**Assumption 4.5** (identifiability over permutation). *Let $[K] = \{1, \ldots, K\}$. For all $k, [K]$ and $\Delta > 0$ we assume: (i) $\arg\min_{k' \neq k} ||\theta_k^{w_q} - \theta_{k'}^{w_p}||^2 > c + \Delta$, and (ii) $||\theta_k^{w_q} - \theta_k^{w_p}||^2 < c$.*

## 4.1 Consistency of algorithm 1

Recall our estimator desiderata are (i) the weak to strong estimator should asymptotically converge to the target (as the number of weak and source samples grows) and (ii) an estimator using one source of data should not asymptotically converge to the target. Under assumptions 2.1 - 2.4 (ii) clearly holds. With no alteration, the source model $P(Y|X)$ clearly does not match $Q(Y|X)$ since $P(k|X) \neq Q(k|X)$. Likewise $Q(Y'|X, k) \neq Q(Y|X, k)$ so long as $\theta_k^{w_q} \neq \theta_k$. We partially establish (i) in the following proposition.

**Proposition 4.6.** *Suppose that the MLE step satisfies the following: For some constants $C > 0$, and $\alpha \in [0, 1]$ it holds that*

$$\mathbb{P}(||\theta - \hat{\theta}||^2 > [\log n_P/n_P]^\alpha) \lesssim exp(-C \log n_P); \quad \theta \in \{\theta_k, \theta_k^{w_p}, \eta_k, \pi_k^p\}_{k=1}^K,$$

$$and \quad \mathbb{P}(||\theta - \hat{\theta}||^2 > [\log n_{Q'}/n_{Q'}]^\alpha) \lesssim exp(-C \log n_{Q'}); \quad \theta \in \{\theta_k^{w_q}, \pi_k^q\}_{k=1}^K.$$

*Then, if $\hat{\theta}_{a(k)}$ denotes the parameter estimates produced by Algorithm 1 and so long as $\Delta > \max\{\mathcal{O}([\log n_P/n_P]^\alpha), \mathcal{O}([\log n_Q/n_{Q'}]^\alpha)\}$ then it holds that*

$$||\hat{\theta}_{a(k)} - \theta_k|| \lesssim \min\{n_P, n_{Q'}\}^{-\frac{\alpha}{2}}.$$

In the case of regression (*cf.* Examples 2.5 and 2.6) we may not be simply interested in estimating the parameters, but actually obtaining the regression function $\mathbb{E}_Q[Y|X]$. Letting $\mu(x; \theta) = \mathbb{E}[Y|X, \theta]$, we may write the function of interest and our estimate $(q(x), \hat{q}(x))$ as

$$q(x) \triangleq \mathbb{E}_Q[Y|x] = \sum_k \frac{\pi_k^q g(x|\eta_k)}{\sum_{k'} \pi_{k'}^q g(x|\eta_{k'})} \mu(x; \theta_k), \quad \hat{q}(x) \triangleq \sum_k \frac{\hat{\pi}_{a(k)}^q g(x|\hat{\eta}_{a(k)})}{[\sum_{k'} \hat{\pi}_{a(k')}^q g(x|\hat{\eta}_{a(k')})]} \mu(x; \hat{\theta}_{a(k)}).$$

**Proposition 4.7.** *Suppose that the MLE step satisfies the conditions in Proposition 4.6. Additionally, suppose that the functions $\log g(x|\eta)$ and $\mu(x|\theta)$ are Lipschitz continuous in their respective parameters. Then (up to logarithmic factors) the estimator $\hat{q}(y|x)$ from Algorithm 1 satisfies*

$$||\hat{q}(x) - q(x)||_{2,Q} \lesssim \min\{n_P, n_{Q'}\}^{-\frac{\alpha}{2}}$$

**Example 4.8.** *When does MLE satisfy the required convergence assumption? For the case of Examples 2.5 and 2.6, suppose that $\varphi(y; x, \theta)$ corresponds to the normal density with mean $\mu(\cdot)$ and variance $\sigma(\cdot)$ parametrized by $x$ and $\theta$. Ho et al. (2022) show that if $\mu(\cdot)$ and $\sigma(\cdot)$ meet a pair of higher-order algebraic independence conditions, then the assumption on MLE holds with $\alpha = 1/4$. We also note that the specification in Example 2.6 meets the needed independence condition.*

*For the more general case, where $g(\cdot)$ is not constant in $x$, possible rates of $\alpha = 1$ or $\alpha = 1/2$ are established in Nguyen et al. (2024b;a).*

## 4.2 Empirical Example

In order to empirically test the findings in the paper, we have adopted a toy empirical setting from Somerstep et al. (2024). The objective is to teach a model a specific persona; we assume that the base model is a mixture of an undesirable and a desirable persona (here the persona corresponds to the concept $k$). We implement Algorithm 1 by clustering data drawn from the source model using K-Means++, computing assignments using cosine similarity, and computing the final estimate by performing supervised fine-tuning on the data from the cluster that is closest to the weak data.

We utilize Dolly as the training set, with weak labels provided by Falcon 7B, Gemma 2B, llama2 7B, and Mistral 8b. GPT-4o-mini acts as the strong model. At test time, we measure the strong model's accuracy and use of the desirable persona on TinyTruthfulQA and TinyAlpaca Eval (Maia Polo et al., 2024). Our findings are consistent with the theoretical findings in the draft. In particular,

1. Weak training biases the model by reducing the accuracy at test time (see content scores of Figure 3)

2. Refinement does not reduce accuracy but does leave Bias by not fully transferring the persona to the model (see style scores of Figure 3)

3. The identification procedure produces a model with no "bias", the persona is learned and the accuracy is not reduced (see both scores of each Figure 3).

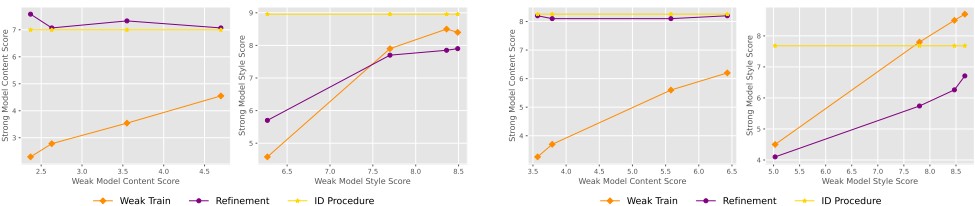

Figure 3: Comparing performance of weak training, refinement and the identification procedure on a persona learning task (TinyTruthfulQA (left) TinyAlpacaEval (right)).

## 5 Conclusion

In this paper, we have studied two popular classes of learning methods for weak-to-strong generalization: (i) weak training and (ii) label refinement within a general latent concept transfer framework, adopted from Somerstep et al. (2024) that includes popular classes of latent concept models for LLMs (Xie et al., 2021; Pathak et al., 2023; Wang et al., 2024). Under the adopted framework, it is shown that there exists a weak-to-strong generalization procedure that produces estimators which asymptotically converge to the target function as the number of unaligned base model and aligned weak model samples grows. This result primarily relies on identification conditions for the aforementioned latent concept models for LLMs. We also show that both weak training and label refinement produce biased estimators of the target function, suggesting that they do not satisfy the consistency property enjoyed by the weak-to-strong generalization algorithm we produce. While both weak training and refinement are tractable, our results suggest the need for further research into weak-to-strong methodologies which enjoy good theoretical properties and remain implementable for practical language modeling problems.

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

# A Proofs

*Proof of Proposition 4.2.* Consider the set up in section 2. We need to show that if $f(x, y; \eta, \theta, K) = f(x, y; \tilde{\eta}, \tilde{\theta}, \tilde{K})$ for almost every $x$ and $y$, then $F(\eta, \theta, K) \equiv F(\tilde{\eta}, \tilde{\theta}, \tilde{K})$. Supposing that the hypothesis holds true, we have

$$\sum_{k=1}^{K} \frac{\pi_k g(x|\eta_k)}{\sum_{k'=1}^{K} \pi_{k'} g(x|\eta_{k'})} \varphi(y; x, \theta_k) = \sum_{k=1}^{\tilde{K}} \frac{\tilde{\pi}_k g(x|\tilde{\eta}_k)}{\sum_{k'=1}^{\tilde{K}} \tilde{\pi}_{k'} g(x|\tilde{\eta}_{k'})} \varphi(y; x, \tilde{\theta}_k) \quad x, y \text{ almost everywhere.}$$

Under the linear independence assumption on $\varphi(y; x\theta)$ if $K \neq \tilde{K}$ then there exists $k^* \in [K]$ such that $\theta_{k^*} \neq \tilde{\theta}_k$ for all $k \in [\tilde{K}]$. But this implies that $g(x|\eta_{k^*}) = 0$ which is a contradiction since $g(x|\eta_{k^*})$ is a probability density. Thus we must have $K = \tilde{K}$. Now note that because the marginal $p(X)$ must remain the same over $f(x; \tilde{\eta}, \tilde{\theta}, K)$ and $f(x; \eta, \theta, K)$, we have that

$$\sum_{k'=1}^{K} \pi_{k'} g(x|\eta_{k'}) - \sum_{k'=1}^{K} \tilde{\pi}_{k'} g(x|\tilde{\eta}_{k'}) = 0$$

Thus is a violation of the linear independence requirement on $g(x; \eta)$ unless we have that the collections $\{g(x|\eta_{k'})\}$ and $\{g(x|\tilde{\eta}_{k'})\}$ are equivalent. Because we are only interested in identifiability over permutation of $k$, WLOG we may take

$$\pi_k g(x|\eta_{k'}) = \tilde{\pi}_k g(x|\tilde{\eta}_{k'})$$

Integrating over $x$ shows that $\pi_k = \tilde{\pi}_k$ which in turn implies that $g(x|\eta_{k'}) = g(x|\tilde{\eta}_{k'})$ so that $\eta_k = \tilde{\eta}_k$. Now putting what we have so far together we get that $\sum_k \pi_k g(x; \eta_k)(\varphi(y; x\theta_k) - \varphi(y; x\tilde{\theta}_k)) = 0$ for almost every $x$. Note that $g(x; \eta_k) \geq 0$ so because there can be no linear relationship among the set $\{\varphi(y; x, \theta_k), \varphi(y; x, \tilde{\theta}_k)\}$, we have that $\theta_k \tilde{\theta}_k$ which establishes that $F_*(\eta, \theta, K) \equiv F_*(\tilde{\eta}, \tilde{\theta}, \tilde{K})$ $\qquad\square$

**Lemma A.1.** *Suppose that $\theta$ and $\theta'$ are distinct transition matrices for a Markov process on a finite state space $\mathcal{H}$ of size $M$. Then there exists a sequence of states $h_{i_1}, \ldots, h_{i_m}$ such that $\mathbf{P}_\theta[h_{i_1} \ldots h_{i_m} h_{i_1}] > \mathbf{P}_{\theta'}[h_{i_1} \ldots h_{i_m} h_{i_1}]$.*

*Proof.* If the matrices $\text{diag}[\theta]$ and $\text{diag}[\theta']$ are not equivalent, then we are finished, as we can take the cycle to be of length 1. If $\text{diag}[\theta] \neq \text{diag}[\theta']$, then there exists states $h_{i_1}, h_{i_2}$ such that $\theta(h_{i_2}|h_{i_1}) > \theta'(h_{i_2}|h_{i_1})$. Now if $\theta(h_{i_1}|h_{i_2}) \geq \theta'(h_{i_1}|h_{i_2})$ we are done, so consider the case where $\theta(h_{i_1}|h_{i_2}) < \theta'(h_{i_1}|h_{i_2})$. In this case, there must exist state $h_{i_3}$ distinct from $h_{i_2}$ and $h_{i_1}$ such that $\theta(h_{i_3}|h_{i_2}) > \theta'(h_{i_3}|h_{i_2})$. Consider the $m' - 1$'th step of this process, where have constructed a sequence of states $h_{i_1}, \ldots h_{i_{m'-1}}$ such that $\theta(h_{i_j}|h_{i_{j-1}}) > \theta'(h_{i_j}|h_{i_j})$ for $1 < j \leq m' - 1$. Suppose $m' - 1 < M$, then either there exists a $j^* < m' - s$ such that $\theta(h_{i_{j^*}}|h_{i_{m'-1}}) \geq \theta'(h_{i_{j^*}}|h_{i_{m'-1}})$ and the proof is complete or there is a distinct (from $\{h_{i_j}\}_{j \leq m'-1}$) state $h_{m'}$ such that $\theta(h_{i_{m'}}|h_{i_{m'-1}}) > \theta'(h_{i_{m'}}|h_{i_{m'-1}})$. Finally, consider the case where have followed this process to the $M$'th step, in particular we are at state $h_{i_M}$, in this case there must exists $h_{i_{j^*}}$ such that $\theta(h_{i_{j^*}}|h_{i_M}) \geq \theta(h_{i_{j^*}}|h_{i_M})$ as otherwise we would have $\sum_j \theta(h_{i_j}|h_{i_M}) < 1$, the existence of such a state completes the proof. $\qquad\square$

*Proof.* We establish the algebraic independence condition for the HMM set up. For a collection of transition matrices $\{\theta_1, \ldots, \theta_k\}$ and prior $(\pi_1, \ldots \pi_K)$ we may write the distribution of the pair $(x, y')$ as

$$p(x, y') = \sum_k \sum_{h \in \mathcal{H}} p(y'|h, \tau^w) p(h|x, \theta_k) p_k(x) \pi_k$$

Recall for identifiability we need that if $\tilde{p}(x, y')$ is the density of $(x, y')$ for the collection of distinct parameters $\{\tilde{\theta}_1, \ldots, \tilde{\theta}_k\}$ and $\{\tilde{\pi}_1, \ldots \tilde{\pi}_K\}$ then $p(x, y')$ and $\tilde{p}(x, y')$ need to be

linearly independent. Note that we may write

$$\alpha p(x, y') + \beta \tilde{p}(x, y') =$$

$$\sum_{h \in \mathcal{H}} p(y'|h, \tau^w) [\alpha [\sum_k p(h|x, \theta_k) p_k(x) \pi_k] + \beta [\sum_k p(h|x, \tilde{\theta}_k) \tilde{\pi}_k]]$$

Given the assumption that there exists a vector of tokens $(o_1^*, \ldots, o_H^*)$ such that $p(o_h^*|h, \tau^w) > 0$ and $p(o_{h'}^*|h, \tau^w) = 0$, the only way for the function to be identically zero is if

$$\alpha [\sum_k p(h|x, \theta_k) p_k(x) \pi_k] + \beta [\sum_k p(h|x, \tilde{\theta}_k) \tilde{p}_k(x) \tilde{\pi}_k] = 0; \quad \text{for all } h \in \mathcal{H}$$

First, we consider the case where the collections $\{\theta_k\}$ and $\{\tilde{\theta}_k\}$ are distinct, WLOG we may say that $\theta_1 \neq \tilde{\theta}_k$ for all $k \in K$, and $\theta_1 \notin \{\theta_k\}_{k>2}$. WLOG, we may reference Lemma *A.1* and note that there exists a cycle $h_{i_1} \ldots h_{i_m} h_{i_1}$ such that $p_1(h_{i_1} \ldots h_{i_m} h_{i_1}) > \tilde{p}_k(h_{i_1} \ldots h_{i_m} h_{i_1})$ for $k \in K$.

By the assumptions, with non-zero probability it may occur that $x = [o_{h_1}^* \ldots o_{h_{i_m}}^*]^l$ for $l \in \mathbb{N}^+$. Letting $p_k \triangleq p_1(h_{i_1} \ldots h_{i_m} h_{i_1})$ for the above to hold, it must necessarily hold that

$$\alpha [\sum_k p(h|h_{i_m}, \theta_k) [\prod_j p(o_{i_j}^*|h_{i_j})]^l [p_k]^l \pi_k] + \beta [\sum_k p(h|h_{i_m}, \tilde{\theta}_k) [\prod_j p(o_{i_j}^*|h_{i_j})]^l [\tilde{p}_k]^l \tilde{\pi}_k] = 0$$

$$\implies \frac{\alpha [\sum_k p(h|h_{i_m}, \theta_k) [p_k]^l \pi_k]}{\beta [\sum_k p(h|h_{i_m}, \tilde{\theta}_k) [\tilde{p}_k]^l \tilde{\pi}_k]} = 1 \text{ for all } l \in \mathbb{N}^+, h \in \mathcal{H}$$

Note though that it is impossible for this to hold for all large $l$, thus if any of the transition matrices are distinct the algebraic independence condition must hold. In the case where the collection of transition matrices $\{\theta_k\}$ are identical $\{\tilde{\theta}_k\}$, a similar argument will show that $\pi_k = \tilde{\pi}_k$ for all $k$ (*i.e.* use the fact that $\theta_k \neq \theta_{k'}$ and select a cycle as we did before). □

*Proof of Proposition 4.6.* The key ingredient is that step 3 of Algorithm 1 produces the correct assignments. By Assumption 4.5 for all $k \in [K]$ and some $\Delta > 0$ we have that (i) $\arg \min_{k' \neq k} ||\theta_k^{w_q} - \theta_{k'}^{w_p}||^2 > c + \Delta$ (ii) $||\theta_k^{w_q} - \theta_k^{w_p}||^2 < c$. Under the assumptions of Proposition 4.6 we have with high probability

$$||\hat{\theta}_k^{w_q} - \hat{\theta}_k^{w_p}||^2 = ||\hat{\theta}_k^{w_q} - \theta_k^{w_q} - (\hat{\theta}_k^{w_p} - \theta_k^{w_p}) + \theta_k^{w_q} - \theta_k^{w_p}||^2$$

$$\leq ||\hat{\theta}_k^{w_q} - \theta_k^{w_q}||^2 + || - (\hat{\theta}_k^{w_p} - \theta_k^{w_p})||^2 + ||\theta_k^{w_q} - \theta_k^{w_p}||^2 \lesssim \mathcal{O}([\log n/n]^\alpha) + c$$

and additionally for $k \neq k'$, we have that

$$||\theta_k^{w_q} - \theta_{k'}^{w_p}||^2 = ||\theta_k^{w_q} - \hat{\theta}_k^{w_q} + (\hat{\theta}_k^{w_p} - \theta_{k'}^{w_p}) - \hat{\theta}_k^{w_q} + \hat{\theta}_{k'}^{w_p}||^2$$

$$\leq ||\theta_k^{w_q} - \hat{\theta}_k^{w_q}||^2 + ||\hat{\theta}_k^{w_p} - \theta_{k'}^{w_p}||^2 + ||\hat{\theta}_k^{w_q} - \hat{\theta}_{k'}^{w_p}||^2$$

$$\lesssim \mathcal{O}([\log n/n]^\alpha) + ||\hat{\theta}_k^{w_q} - \hat{\theta}_{k'}^{w_p}||^2$$

Or re-arranging the inequality

$$||\hat{\theta}_k^{w_q} - \hat{\theta}_{k'}^{w_p}||^2 \geq ||\theta_k^{w_q} - \theta_{k'}^{w_p}||^2 - \mathcal{O}([\log n/n]^\alpha) \geq c + \Delta - \mathcal{O}([\log n/n]^\alpha)$$

so that for if $\Delta > \mathcal{O}([\log n/n]^\alpha)$ it must hold true that $\arg \min_{k'} ||\theta_k^{w_q} - \theta_{k'}^{w_p}||^2 = k$. □

*Proof of Proposition 4.7.* In the notation of Algorithm 1 we have established that

$$\hat{q}(x)| \leftarrow \sum_k \frac{\hat{\pi}_{a(k)}^q g(x|\hat{\eta}_{a(k)})}{[\sum_{k'} \hat{\pi}_{a(k')}^q g(x|\hat{\eta}_{a(k')})]} \mu(x; \hat{\theta}_{a(k)})$$

$$a(k) = k \text{ for all } k \in [K]$$

So we seek a bound on

$$\int_{\mathcal{X}} || \sum_k \frac{\hat{\pi}_k^q g(x|\hat{\eta}_k)}{[\sum_{k'} \hat{\pi}_{k'}^q g(x|\hat{\eta}_{k'})]} \mu(x;\hat{\theta}_k) - \sum_k \frac{\pi_k^q g(x|\eta_k)}{[\sum_{k'} \pi_{k'}^q g(x|\eta_{k'})]} \mu(x;\theta_k) ||_2^2 dQ(x)$$

$$\text{Triangle Inequality} \overset{\leq}{=} \sum_k \int_{\mathcal{X}} || \frac{\hat{\pi}_k^q g(x|\hat{\eta}_k)}{[\sum_{k'} \hat{\pi}_{k'}^q g(x|\hat{\eta}_{k'})]} \mu(x;\hat{\theta}_k) - \frac{\pi_k^q g(x|\eta_k)}{[\sum_{k'} \pi_{k'}^q g(x|\eta_{k'})]} \mu(x;\theta_k) ||_2^2 dQ(x)$$

So what remains is to establish the Lipschitz continuity of the function $\mu_Q(x;\pi,\eta,\theta) = \frac{\pi_k^q g(x|\eta_k)}{[\sum_{k'} \pi_{k'}^q g(x|\eta_{k'})]} \mu(x;\theta_k)$. Note that we can equivalently write

$$\mu_Q(x;\pi,\eta,\theta) = \text{SFTMAX}([\log \pi_1 g(x|\eta_1), \ldots, \log \pi_K g(x|\eta_K)])_k \mu(x|\theta_k)$$

where SFTMAX denotes the softmax function. Thus Lipschitness follows from the assumption that $\log \pi g(x|\eta)$ and $\mu(x|\theta_k)$ are both Lipschitz and bounded and the fact that the softmax function composed with a set of $K$ Lipschitz function is itself Lipschitz. □

*Proof.* First, multiply the optimization problem by $1/1+\lambda$

$$\arg\min_{\theta \in \Theta} \frac{1}{1+\lambda} \sum_{i=1}^{n_p} (f(x_i,\theta) - y_i)^2 + \frac{\lambda}{1+\lambda} \sum_{i=1}^{n_{Q'}} (f(x_i,\theta) - y_i')^2$$

Then let $\eta = \frac{1}{1+\lambda}$ and expand terms to get

$$\arg\min_{\theta \in \Theta} \sum_{i=1}^{n} f(x_i,\theta)^2 - 2\eta y_i f(x_i,\theta) - 2(1-\eta)y_i' f(x_i\theta) + \eta y_i^2 + (1-\eta)y_i'^2$$

Next we add and subtract $(\eta y_i + (1-\eta)y_i')^2$ to complete the square, and drop the addition of any terms that are not a function of $\theta$.

$$\hat{\theta}_{\text{wt}} = \arg\min_{\theta \in \Theta} \sum_{i=1}^{n} (f(x_i,\theta) - (\eta y_i + (1-\eta)y_i'))^2$$

Due to the assumption that the function $f(x;\theta)$ is Lipschitz in $\theta$, the empirical loss satisfies uniform convergence and thus

$$\hat{\theta}_{\text{wt}} \overset{p}{\to} \arg\min_{\theta} \mathbb{E}_{x,y,y'}(f(x,\theta) - (\eta y + (1-\eta)y'))^2$$

Which implies that

$$\hat{\theta}_{\text{wt}} \overset{p}{\to} \arg\min_{\theta} \mathbb{E}_x(f(x,\theta) - (\eta \sum_k \pi_k^p f(x;\theta_k) + (1-\eta) \sum_k \pi_k^q f(x;\theta_k^{qw})))^2$$

Note that we may write

$$\mathbb{E}_X(\mathbb{E}_y[y|x] - (\eta \sum_k \pi_k^p f(x;\theta_k) + (1-\eta) \sum_k \pi_k^q f(x;\theta_k^{qw})))^2$$

$$= \eta ||\epsilon_P||^2 + (1-\eta)^2 ||\epsilon_{Q'}||^2 + \eta(1-\eta)\epsilon_P^T \epsilon_{Q'}$$

By the assumption on $\Theta$, $f(x;\theta_{\text{wt}}^*)$ must be within a ball of radius $r^2$ of $\eta \sum_k \pi_k^p f(x;\theta_k) + (1-\eta) \sum_k \pi_k^q f(x;\theta_k^{qw})$, which completes the proof.

□

*Proof of Proposition 3.2.* We have

$$\int_{y'} p(y|x,y')q(y'|x)dy' = \int_{y'} p(y|x,y') \sum_{k'} q(y'|x,k')q(k'|x)dy'$$

$$= \int_{y'} [\sum_k p(y|x,y',k)p(k|x,y')] \sum_{k'} q(y'|x,k')q(k'|x)dy$$

$$= \sum_k p(y|x,k) \int_{y'} p(k|x,y') \sum_{k'} q(y'|x,k')q(k'|x)dy'$$

$$= \sum_k p(y|x,k)[\sum_{k'} q(k'|x) \int_{y'} p(k|x,y')q(y'|x,k')dy'$$

$$\square$$

*Calculation of Example 3.4.* By Proposition 3.2 we have

$$\hat{q}(y|x) = \sum_k p(y|x,k)\hat{q}(k|x),$$

$$\hat{q}(k|x) = \int_{y'} p(k \mid x,y')q(y'|x,k_Q)dy'$$

Now by Bayes' rule note that

$$\int_{y'} p(k \mid x,y')q(y'|x,k_Q)dy'$$

$$= p(k|x) \int_{y'} \frac{p(y' \mid x,k)q(y'|x,k_Q)}{\sum_k p(k|x)p(y'|x,k)} dy'$$

$$\leq p(k|x) \int_{y'} \frac{p(y' \mid x,k)q(y'|x,k_Q)}{p(k|x)p(y'|x,k) + p(k_Q|x)p(y'|x,k_Q)}$$

Now we plug in the parametric form of Assumption 2.4 to arrive at

$$q(\hat{k}|x) \leq p(k|x) \times \int_{-\infty}^{\infty} \frac{e^{-\frac{1}{2\sigma^2}(y'-\mu(x;\theta_k^{wp}))^2} e^{-\frac{1}{2\sigma^2}(y'-\mu(x;\theta_{k_Q}^w))^2}}{e^{-\frac{1}{2\sigma^2}(y'-\mu(x;\theta_k^{wp}))^2} + e^{-\frac{1}{2\sigma^2}(y'-\mu(x;\theta_{k_Q}^w))^2}} dy'$$

Multiplying the numerator and denominator by $e^{\frac{1}{2\sigma^2}(y'-\mu(x;\theta_{k_Q}^w))^2}$, and letting $\Delta_k^2(x)$ be defined as in the statement we have

$$\hat{q}(k|x) \leq p(k|x) \times \int \frac{e^{-\frac{1}{2\sigma^2}(y'-\mu(x;\theta_{k_Q}^w))^2}}{1 + e^{\Delta_k^2(x)-2\Delta_k(x)(y-\mu(x;\theta_{k_Q}^w))}} dy'$$

Then we simply break the integral into the two cases: $|y' - \mu(x;\theta_Q^w)| < |\Delta_k(x)|/4$ and $|y' - \mu(x;\theta_Q^w)| \geq |\Delta_k(x)|/4$ to arrive at the final result. $\square$

## B Details of Figure 1

Figure 1 tests three weak to strong methods using the GSM8K (Cobbe et al., 2021) data set. Weak labels are produced by Llama-2-7B-Chat (Touvron et al., 2023), Mistral-7B (Jiang et al., 2023), and Gemma-1.2B (Team and Others, 2024) models. To provide some expertise on the task, each weak model recieved supervised fine tuning with gold standard data, the data is produced by (Yang et al., 2024b). The strong model is GPT-3.5-Turbo-0125 (Achiam et al., 2024). We compare thee weak to strong methods: (i) simply training GPT-3.5-Turbo on the weak data (ii) Using the ICL refinement method of Somerstep et al. (2024) (iii) an oracle method where GPT-4o produces answers to the training set that GPT-3.5-Turbo is trained on. Evaluation is done on a provided test set with answer key included, GPT-4o is used to judge if the given test response matches the answer key in both the reasoning and the final answer.

