# OpenReview forum: "Limitations of refinement methods for weak to strong generalization"
_colmweb.org/COLM/2025/Conference — COLM 2025_

### Official Review · Reviewer_PyrR · 2025-05-11

**Rating:** 7
**Confidence:** 4
**Ethics Flag:** 1

**Summary:**

The paper is on weak to strong generalization, where a smaller model (weak) provides data to train a larger model (strong). The authors argue that existing methods like label refinement and weak training have irreducible errors and propose a deconvolution-based method that might perform better but isn't practical yet. their proposed method only highlights a theoretical performance gap.

**Questions To Authors:**

Propositions 4.1 and 4.3 rely on algebraic independence and anchor words. How frequently do these conditions hold in practice for modern LLMs? Have you empirically validated these assumptions (e.g., in HMMs or regression settings), or are they primarily theoretical?

The latent concept shift framework assumes shifts only occur in KK and Y′Y′, while XX and Y∣X,KY∣X,K remain unchanged. How do you justify this assumption for real-world LLM alignment, where shifts in XX or Y∣XY∣X (e.g., due to time, topic, distribution shifts, or human preferences etc) might also occur?

**Reasons To Accept:**

The paper has a potential theoretical and analytical contribution to weak-to-strong generalization in LLM alignment. This ** may ** trigger some insights for practical explorations along this direction. However, have to note that this as more of analytical contributions, as additional data points for researcher to examine. The assumptions of the theoretical aspect are in wave-of-hand style.

**Reasons To Reject:**

[1] limited practical impact and insufficient empirical validation.
[2] the empirical evaluation is narrowly focused on GSM8K with sparse experimental details
[3] only incremental analytical value

---

> ### Author Response · Authors · 2025-05-31
> **Response to Reviewer PyrR**
>
> Thank you for taking the time to make helpful comments on our work; below, we address specific questions.
>
> *Propositions 4.1 and 4.3 rely on algebraic independence and anchor words. How frequently do these conditions hold in practice for modern LLMs? Have you empirically validated these assumptions (e.g., in HMMs or regression settings), or are they primarily theoretical?*
>
> In order to better connect our theoretical discussion to practical settings, we have added a pseudo-implementation of Algorithm 1 to a toy empirical setting from [1] that utilizes text data and LLMs. The objective is to teach a model a specific persona; we assume that the base model is a mixture of an undesirable and desirable persona (here the persona corresponds to the concept $k$).
>
> The experiment we ran generally meets most of these conditions. Anchor tokens/words are present for all of the personas (e.g. 'Ahoy' for a pirate). Additionally, the clustering step largely correctly groups the different personas, suggesting that the different components meet some identifiability criteria. The success of the identification procedure at this task demonstrates our finding that if these criteria are met, further improvements on refinement and weak training are possible. All figures can be found at (https://anonymous.4open.science/r/COLMRebuttal-3DF4).
>
> *The latent concept shift framework assumes shifts only occur in KK and Y′Y′, while XX and Y∣X,KY∣X,K remain unchanged.*
>
> Weak to strong generalization is a problem setting meant to emulate a human teaching a superhuman model a new task. The assumption that $P_X$ remains unchanged is essentially assuming that the human has access to the type of problem it wishes to teach the model, one that seems reasonable in this context. The assumption that $P_{Y|X, K}$ remains unchanged while $P_K$ does change is tapping into the idea that the strong model is "superhuman", or at the very least is quite capable. $P_{Y|X, K} = Q_{Y|X, K}$ encodes this, if the latent concept is correct then the strong model produces good outputs.
>
> We agree that shifts in P_X and P_Y|X can easily happen in LLM settings, and would be interesting directions to study in the future.
>
> [1] S. Somerstep et al. A Transfer Learning Framework for Weak to Strong Generalization. ICLR 2025

---

### Official Review · Reviewer_BFqD · 2025-05-11

**Rating:** 6
**Confidence:** 4
**Ethics Flag:** 1

**Summary:**

This paper develops a latent‑concept‑shift framework to analyze weak‑to‑strong generalization (WSG) techniques for aligning large language models. Within this probabilistic mixture‑of‑experts setting, the authors (i) prove that weak training and label refinement are inconsistent: each suffers irreducible bias that prevents convergence to the target distribution; (ii) derive information‑theoretic lower bounds that expose a performance gap between these methods and an oracle; and (iii) construct an identification‑based two‑stage procedure (Algorithm 1) that is provably consistent whenever mixture components are identifiable and the weak/strong mixtures are statistically distinguishable. They further supply sufficient algebraic‑independence conditions covering regression and HMM instantiations, and replicate recent empirical results on GSM8K to illustrate the theory.

**Questions To Authors:**

Some proofs (Appendix A) rely on lengthy algebraic‑independence arguments that could be condensed; the paper would benefit from a running toy example to ground intuition.

**Reasons To Accept:**

1. Recasts WSG as a transfer‑learning problem with latent concept drift, unifying several prior toy models under one general graphical framework.
2. Shows there is a statistically consistent solution (Algorithm 1) and precisely states the identifiability requirements under which it works.

**Reasons To Reject:**

1. Algorithm 1 requires full‑batch MLE over latent mixtures (often NP‑hard) and presumes the number of concepts K is known; no scalable approximation or empirical test is provided.
2. Key identifiability conditions (anchor tokens, unique mixture components, observed weak labels in the source domain) may rarely hold for real LLMs. Results may therefore overstate attainable guarantees.
3. Apart from a single GSM8K replication, there is no quantitative study showing the proposed estimator working in practice, nor any ablation on assumption violations.
4. Empirical or theoretical contrasts with more recent WSG variants (e.g., dynamic logit fusion, preference optimisation) are missing.

---

> ### Author Response · Authors · 2025-05-31
> **Response to reviewer BFqD**
>
> Thank you for taking the time to make helpful comments on our work; below, we address specific questions.
>
> *Addressing scalability of Algorithm 1*
>
> We agree that there are examples of LLM tasks where Algorithm 1 is likely not useful; it is more meant to conceptually demonstrate that within a framework used to study weak to strong generalization, it is theoretically possible to do better than currently proposed methods. That said, we do believe that an empirical example would help drive this point home. Below we discuss one that we will include in the paper.
>
> *Addressing need for empirical validation of findings*
>
> We have adapted a toy empirical setting from [1] that utilizes text data and LLMs. The objective is to teach a model a specific persona; we assume that the base model is a mixture of an undesirable and desirable persona (here the persona corresponds to the concept $k$). We implement Algorithm 1 by clustering data drawn from the source model using K-Means++, computing assignments using cosine similarity, and computing the final estimate by performing supervised fine-tuning on the data from the cluster that is closest to the weak data.
>
> In the linked figure, we score how refinement, weak training, and Algorithm 1 perform at teaching the persona and allowing the stronger model to maintain accuracy. We see that weak training can teach the performance but reduces accuracy, while refinement does not reduce accuracy but struggles to fully transfer the persona, and Algorithm 1 is generally successful at both. All figures can be found at (https://anonymous.4open.science/r/COLMRebuttal-3DF4).
>
> *Addressing Key identifiability conditions (anchor tokens, unique mixture components, observed weak labels in the source domain)*
>
> The experiment we ran generally meets most of these conditions. Anchor tokens are present for all of the personas (e.g. 'Ahoy' for a pirate). Providing that each of the personas is distinct, and the personas correspond to the mixture components' unique mixture components. Note that in this case, we do not use any weak labels from the source but as we comment in the paper "we can instead opt to utilize θk in the assignment step," which we do here.
>
> [1] S. Somerstep et al. A Transfer Learning Framework for Weak to Strong Generalization. ICLR 2025

---

### Official Review · Reviewer_nFeH · 2025-05-12

**Rating:** 6
**Confidence:** 2
**Ethics Flag:** 1

**Summary:**

Authors investigate two primary methods of weak-to-strong (w2s) alignment: refinement and weak-to-strong training. The motivation starts from the performance gap between Oracle training and W2S methods. Specifically, refinement works better than weak training but worse than oracle training. Authors then show that the two methods produce biased estimators (though convergence) and label refinement has irreducible error. To this end, authors propose a theoretical method to leverage the problem.

**Questions To Authors:**

Typo:
* Extra bracket in formula on page 4 (top)
* Misalignment the formula on page 5 (bottom)

**Reasons To Accept:**

* The angle of transfer learning is interesting, which provides a new perspective in understanding weak-to-strong
* Examples are helpful for understanding
* Clear structure

**Reasons To Reject:**

I am not an expert in theory and transfer learning, so I did not dive into the details of the theory much. Instead, I have some general concerns:

* The key assumption is latent concept transfer. However, I do not quite understand why this assumption is selected to analyze the weak-to-strong alignment instead of other assumptions. For example, how good is assumption 2.4 in describing the real w2s setting?

* How can the theoretical method become practical, considering g() and \phi are unknown?

---

> ### Author Response · Authors · 2025-05-31
> **Response to Reviewer nFeH**
>
> Thank you for taking the time to make helpful comments on our work; below, we address specific questions.
>
> *I do not quite understand why the latent concept assumption is selected to analyze the weak-to-strong alignment instead of other assumptions. For example, how good is assumption 2.4 in describing the real w2s setting?*
>
> We opted to study the latent concept setting as it is of interest in the weak to strong literature ([1]), incorporates popular toy models for LLMs ([2], [3], [4]) and is used as a theoretical justification for refinement techniques ([1], [5]). We do agree that there are other interesting and valid ways to model weak to strong generalization, but we hope that these examples demonstrate that the latent concept setting is a reasonable choice.
>
> *How can the theoretical method become practical, considering g() and \phi are unknown?*
>
> We agree that there are examples of LLM tasks where Algorithm 1 is likely not useful; it is more meant to conceptually demonstrate that within a framework used to study weak to strong generalization, it is theoretically possible to do better than currently proposed methods.
>
> That said, to better demonstrate this point we have adapted a toy empirical setting from [1] that utilizes text data and LLMs. The objective is to teach a model a specific persona, we assume that the base model is a mixture of an undesirable and desirable persona (here the persona corresponds to the concept $k$). We implement Algorithm 1 by clustering data drawn from the source model using K-Means++, computing assignments using cosine similarity, and computing the final estimate by performing supervised fine-tuning on the data from the cluster that is closest to the weak data.
>
> In the linked figure, we score how refinement, weak training, and Algorithm 1 perform at teaching the persona and allowing the stronger model to maintain accuracy. We see that weak training can teach the performance but reduces accuracy, while refinement does not reduce accuracy but struggles to fully transfer the persona, and Algorithm 1 is generally successful at both. All figures can be found at (https://anonymous.4open.science/r/COLMRebuttal-3DF4).
>
> [1] S. Somerstep et al. A Transfer Learning Framework for Weak to Strong Generalization. ICLR 2025
>
> [2]. Xie et al. An Explanation of In-context Learning as Implicit Bayesian Inference. ICLR 2022.
>
> [3]. Pathak et al. Transformers can optimally learn regression mixture models. ICLR 2024.
>
> [4] Wang et al. Large language models are latent variable models: Explaining and finding good demonstrations
> for in-context learning. NeurIPS 2024.
>
> [5]. Yang et al. Weak-to-strong reasoning. 2024.

---

> > ### Comment · Reviewer_nFeH · 2025-06-10
> >
> > Thanks, authors, for providing a rebuttal. I read the rebuttal and decided to maintain my tendency to accept at a mild level.

---

### Official Review · Reviewer_PQE1 · 2025-05-13

**Rating:** 6
**Confidence:** 3
**Ethics Flag:** 1

**Summary:**

The paper studies methods that try to turn weak supervision into strong performance in the so-called weak-to-strong generalization setting. It places both weak training and label refinement inside a latent concept transfer framework, proves that these common approaches carry an irreducible bias, and gives a constructive though impractical two-step procedure that can close the gap under identifiable mixture assumptions. The analysis is mathematically careful and unifies several prior toy models in one general view.

**Questions To Authors:**

Can you provide a small synthetic experiment that illustrates the latent concept identification algorithm and shows the predicted consistency gap?

**Reasons To Accept:**

The topic is timely because many groups now rely on small helper models to align larger ones. The theoretical framework is broader than earlier papers and includes mixture of experts, hidden Markov mixtures, and regression mixtures in one shot. Proofs are detailed and the key identifiability condition is clearly spelled out.

**Reasons To Reject:**

The contribution is purely theoretical and the assumptions needed for the positive result are strong, in particular anchor-word style conditions and full knowledge of weak labels in the source domain. No empirical study shows whether the bias predicted by the theory matters in realistic systems. In addition, the exposition is dense and some proofs rely on lengthy technical lemmas that may hide important edge cases.

---

> ### Author Response · Authors · 2025-05-31
> **Response to reviewer PQE1**
>
> Thank you for taking the time to make helpful comments on our work; below, we address specific questions.
>
> *Can you provide a small synthetic experiment that illustrates the latent concept identification algorithm and shows the predicted consistency gap?*
>
> In lieu of a synthetic experiment, we have adapted a toy empirical setting from [1] that utilizes text data and LLMs. The objective is to teach a model a specific persona, we assume that the base model is a mixture of an undesirable and desirable persona (here the persona corresponds to the concept $k$). We implement Algorithm 1 by clustering data drawn from the source model using K-Means++, computing assignments using cosine similarity, and computing the final estimate by performing supervised fine-tuning on the data from the cluster that is closest to the weak data.
>
> In the linked figure, we score how refinement, weak training, and Algorithm 1 perform at teaching the persona and allowing the stronger model to maintain accuracy. We see that weak training can teach the performance but reduces accuracy, while refinement does not reduce accuracy but struggles to fully transfer the persona, and Algorithm 1 is generally successful at both. All figures can be found at (https://anonymous.4open.science/r/COLMRebuttal-3DF4).
>
> We also mention that in this toy setting it is easy to identify anchor words. For example, the word "Ahoy" is only used if the persona is pirate, and "Hark" is only used if the persona is knight.
>
> [1] S. Somerstep et al. A Transfer Learning Framework for Weak to Strong Generalization. ICLR 2025

---

> > ### Comment · Reviewer_PQE1 · 2025-06-05
> >
> > Thank you for your response, and for an interesting work!

---

### Author Response · Authors · 2025-05-31
**Response to reviewers**

Thank you to tall the reviewers for your time in providing thoughtful reviews of our work!

As most reviewers requested some form of empirical test of the findings in the paper, we have adapted a toy empirical setting from [1] that utilizes text data and LLMs. The objective is to teach a model a specific persona; we assume that the base model is a mixture of an undesirable and desirable persona (here the persona corresponds to the concept $k$). We implement Algorithm 1 by clustering data drawn from the source model using K-Means++, computing assignments using cosine similarity, and computing the final estimate by performing supervised fine-tuning on the data from the cluster that is closest to the weak data.

We utilize Dolly as the training set, with weak labels provided by Falcon 7B, Gemma 2B, llama2 7B, and Mistral 8b. GPT-4o-mini acts as the strong model. At test time, we measure the strong model's accuracy and use of the desirable persona on TruthfulQA and Alpaca Eval. Our findings our consistent with the theoretical findings in the draft.

* Weak training biases the model by reducing the accuracy at test time (see content scores of Figures)
* Refinement does not reduce accuracy but does leave Bias by not fully transferring the persona to the model (see style scores of Figures)
* The identification procedure produces a model with no ``bias", the persona is learned and the accuracy is not reduced (see both scores of each Figure).

All figures can be found at (https://anonymous.4open.science/r/COLMRebuttal-3DF4).

[1] S. Somerstep et al. A Transfer Learning Framework for Weak to Strong Generalization. ICLR 2025

---

### Decision · Program_Chairs · 2025-07-08

**Decision:**

Accept

**Comment:**

The paper provides a valuable theoretical contribution to the problem of weak-to-strong generalization. The reviewers were in consensus about the paper's novelty and the importance of its theoretical perspective. The primary weakness identified by reviewers was the initial lack of empirical validation for the theoretical claims. The authors addressed this during the rebuttal by adding a toy experiment that illustrates the predicted performance gaps and shows their proposed algorithm working in a controlled setting.

For the camera-ready version, I would recommend that the authors add the new experiments from the rebuttal into the main paper, as they strengthen the work.

Overall, the paper's novel framework and insightful analysis of a critical problem in LLM alignment make it a solid contribution to the conference.